# Complex Network Analysis of Mass Violation, Specifically Mass Killing

**DOI:** 10.3390/e24081017

**Published:** 2022-07-23

**Authors:** Iqra Erum, Rauf Ahmed Shams Malick, Ghufran Ahmed, Hocine Cherifi

**Affiliations:** 1School of Computing, National University of Computer and Emerging Sciences (NUCES), Karachi 75270, Pakistan; k173055@nu.edu.pk (I.E.); gahmad78@gmail.com (G.A.); 2Laboratoire Electronique, Informatique et Image (Le2i) UMR 6306 CNRS, Université de Bourgogne, 21078 Dijon, France; hocine.cherifi@gmail.com

**Keywords:** complex networks, Mass Killing, social network analysis

## Abstract

News reports in media contain news about society’s social and political conditions. With the help of publicly available digital datasets of events, it is possible to study a complex network of mass violations, i.e., Mass Killings. Multiple approaches have been applied to bring essential insights into the events and involved actors. Power law distribution behavior finds in the tail of actor mention, co-actor mention, and actor degree tells us about the dominant behavior of influential actors that grows their network with time. The United States, France, Israel, and a few other countries have been identified as major players in the propagation of Mass Killing throughout the past 20 years. It is demonstrated that targeting the removal of influential actors may stop the spreading of such conflicting events and help policymakers and organizations. This paper aims to identify and formulate the conflicts with the actor’s perspective at a global level for a period of time. This process is a generalization to be applied to any level of news, i.e., it is not restricted to only the global level.

## 1. Introduction

Complex network analysis has caught enough attention of researchers in multiple fields due to its power to uncover complex mechanisms, structures, and dynamics [1,2,3], spanning from social sciences to biological sciences. Network analysis is the investigation of some measurements that depict the structure of a system or catch parts of people’s position in the system [4,5]. Studying the perspective that network analysis shows is necessary to understand the interlinked world [6]. Today, with the increase in activity, social dynamics are being studied quantitatively and qualitatively [2,4,7,8]. This multidisciplinary research area has been a traditional ground truth for social scientists, but now, there is also the extensive participation of physicists, computer scientists, mathematicians, and others. Theories of social sciences using tools for statistical physics, along with multiple science disciplines such as mathematics, statistics, applied physics, and computer science, are known as computational social sciences. As the name suggests, it uses computational approaches to deal with social affairs. The study of social behavior is fascinating as it shows the behavior of interacting agents/actors.

Social behavior of a society can be both constructive (positive, e.g., building society or shaping culture) or destructive (negative, e.g., conflicts, wars, or battles) at several temporal and spatial scales. Negative interactions occur due to social, economic, and political pre-conditions.

This paper works on the Mass Killing dataset. The genocide scholar Ervin Staub uses the term “Mass Killing” to describe the Killing of a group of people without the intention of eliminating the whole group or its representatives [9]. In the reference section of the encyclopedia (1999), Israel Charny defines generic genocide as “the mass executing of generous quantities of people, when not throughout military activity against the military powers of a declared foe, under states of the basic lack of protection and weakness of the people in question“ [10]. Valentino uses the term ”Mass Killing” and appropriately characterizes it as “the deliberate murdering of countless noncombatants”. The word “noncombatants” recognizes mass slaughtering from fight passing in war, which happens as soldiers battle against one another. The “enormous number” he chooses as the edge to mass-murdering seems to be “in any event fifty thousand purposeful passing’s through the span of five or fewer years, which average in any event 10,000 executed every year” [11]. The availability of many digital media news datasets has made it possible to analyze networks and bring critical insight into the individuals/actors/organizations/nations involved.

Mass violation and Mass Killing is a significant issue affecting people worldwide (Burma, Syria, Ukraine, Yemen, and in different forms in developed countries). It is getting worse day by day. Understanding the structural composition of events is essential to predicting such events in advance. Furthermore, understanding the contributing actors along with the interactions at the micro and macro level may allow policy makers, strategists, and decision makers to deal with evolving events in their early days. Several attempts are made to understand and model such events as mass shootings [12,13] and cyber violence [14] using agent-based modeling techniques. In [15], the authors conducted a comparative analysis of foiled and completed mass shootings in the United States between 2000 and 2019.

We do not have any tool, model, or pathway that we can use to analyze such news occurrences and draw meaningful conclusions structurally. To understand the interactions of actors, the development of mass violence events, and their evolutionary dynamics, it is vital to have considerably reliable data in temporal settings. Earlier, the absence of such data restricted social scientists from understanding and modeling such events with structural representations. However, with the availability of data on social media and the abundance of news data, scientists can now understand the dynamics of such events. Indeed, studies understanding the structural representations of such events are receiving attention from the scientific community. However, to the best of our knowledge, this is one of the early attempts to employ news data to represent events of mass violence in a structure, particularly with the help of complex network tools. Several recent studies reported data-driven models and frameworks to study mass violence, including [16,17]. Public response data can also be employed to model the violence at the mass level [18]. However, comparatively, it is challenging to consider social media and news media as a structured sources of information that can be credible and robust for such studies. With the development of the Global Database of Events, Language and Tone (GDELT) service, the robustness and availability of data are no longer a big challenge. To conduct such a study, obtaining credible data over a long period was a significant challenge, as many sources do not supply such accurate and dependable data. Apart from GDELT, a few other sources https://acleddata.com/#/dashboard, accessed on 24 April 2021, and https://harvard.edu/dataverse/icews, accessed on 24 April 2021 provide similar events-based news datasets. However, the GDELT update period for its the data set is much shorter than the others, i.e., every 15 min, allowing for a more global view.

As a result, GDELT offers an excellent opportunity to explore such events based on actual news, allowing us to look into structural patterns based on three measures: actors, co-actor pairs, and actor degree, all of which occur in various parts of the world. To the best of our knowledge, no theory has been created on GDLET data to examine Mass Killing and discover its structural representation.

In the present study, we address a few compelling questions, including:Is there any pattern to all the Mass Killing incidents that occur worldwide revealed through the network’s structural organization?If structural patterns exist, is there some resolution based on those structural attitudes?If the structure is recurrent, it is recurrent in news stories of Mass Killings?Is there any structural coherence or pattern between occurrences?

Indeed, giving a positive answer to those questions indicates that there is anything in the resolution we can reuse in other events. Consequently, such structural representations can help us perform necessary actions in the early days of events to prevent massive destruction to humanity.

This paper aims to identify and formulate a process that can help investigate time-variant conflicting data, exploring it through actors involved in such events. This process has been applied to the Global Database of Events, Languages, and Tone (GDELT), a global news collection. However, it is not limited to this case only. Indeed, one can use it with news having power-law properties at any level in their data.

## 2. Related Work

This work applies different approaches for investigating actor, co-actor pair, and actor connectivity with others. Therefore, the literature review consists of work related to mass violence and the steps involved in this work, applied by other authors to solve their issues.

In previous decades, the investigation of complex systems has been a fast-growing research area. In 1950, two scientists discovered complex network topology by random graph theory. As complex networks have many nodes with many interconnections, they cannot be understood easily by visual inspection. Complex networks display complex behavior. Therefore, it is essential to inspect their microscopic details to analyze them in-depth. Various terminologies related to complex networks are described [19]. Network analysis has been used in many real-world phenomena on various networks, e.g., on US migration networks data [7], conflicting data [4,5], human life/interactions [20,21], railway express network [22], ecological networks [23,24], social media network data [25,26], and biological network data [27,28,29]. Analyzing a network through multiple approaches such as quantitative and qualitative ones can provide a much more detailed explanation, deeper insight, and better understanding of the development of the network, network dynamics, and growth pattern of nodes. A step-by-step guide to designing a mixed-method approach has been explained. The mixed-method approach (quantitative and qualitative) has become popular as it provides a deeper understanding of the processes [30]. The mixed-method approach to social network analysis can provide an important understanding of analytical and network dynamics [8]. Mass violence is experimentally infrequent. Examining mass violence presents various methodological challenges. The complex idea of mass violence occasions, which may have sprouted years earlier, makes investigators utilize regular examination strategies risky. Complexity sciences and the interdisciplinary field of computational social science offer new logical ideal models, implementations, and tools appropriate to the investigation of complicated and dynamic occurrences such as mass violence [31].

Network analysis has been reported on data related to mass violence, e.g., mass violation, Mass Killing, murdering. Three types of computational models have been presented that will probably be specifically related to mass violence and incentive to the danger appraisal and managing community [31]. Murder by structure has been analyzed in which neighborhood social establishments and on-screen characters are reliant; that is, the social systems of people and gatherings (regardless of whether prosocial or degenerate) reach beyond neighborhood limits and encourage the spatial contagion of murder. Specifically, some freak and criminal practices are particularly infectious, such as medicate management, betting rings, and group practices [32].

Tasks of social network analysis include [33] the identification of the most influential, important, dominant, effective node or nodes [34,35], visualization of the interaction between nodes [36], group analysis [37,38], network resilience [39,40], information spreading [41,42], disease immunization [43,44], conflicts [4,5], measuring and visualizing the growth of nodes, i.e., calculating network growth [45]. Any network data representing the structural relationship of entities is called a graph. The graph is the interaction of components known as nodes, and the component that links nodes with each other is called an edge link or the actor pair link [33]. There are different graph types: directed, weighted, unweighted, undirected graphs, multigraphs, etc. The dataset that is the topic of this study is a multigraph. Multigraphs are graphs or networks where multiple edges are allowed between nodes, and self-loop is also allowed [46]. Multigraphs are also called multivariate networks. In a multivariate network, there are various relations where everyone can be viewed as a binary variable factor in the group of actors comprising a univariate network [47,48].

Power laws are well-known all through nature, in areas such as astronomy, etymology, and neuroscience [20,27,29,49]. The power-law distribution is hypothetically fascinating due to the heavy tail. Whenever a researcher wants to study data distribution to know whether power law is an appropriate distribution, then there are a few questions one needs to answer, which are very well highlighted in [50]. Investigations of empirical distribution that keep a power law generally estimate the scaling parameter α. There are a few different ways of assessing it, but of course, not all will give accurate results and they are all biased. A literature review shows that the Maximum Likelihood estimate (MLE) is the best identified method. Indeed, it is reliable and unbiased and has been used by authors frequently [34,50,51,52,53]. Exponents of power-law are critical as they provide inference about the underlying phenomena, so this should be estimated very carefully.

The scaling boundary ordinarily lies in the range 2 < α < 3, even though there are infrequent exceptional cases. Fitting a power-law conveyance to exact information, just as estimating the power-law exponents of that fit, is non-trivial. The most significant primary attributes in investigating large scale-free complex networks in empirical distribution are degree distributions [49]. One often favors Complementary Cumulative Distribution Functions (CCDF) for visualizing a heavy tail distribution.

Complex network analysis for the growth of the citation of scientific papers has been studied. In particular, building up a stochastic model of reference elements is dependent on copying the redirection-triadic closure mechanism. In a corresponding, more lucid way, the model looks both for measurements of references of scientific papers and for their elements’ dynamics [2].

The identification of influencers is a hot topic in social network analysis. Many centrality measures have been proposed to address this problem [54,55]. In [34], the authors propose a method of local information dimensionality (LID) to identify the centrality of actors. It relies on the local structural properties around the node. The locality scale is proportional to the maximum shortest distance from the node. The Shannon entropy measures the information contained in boxes with increasing size. The higher the local dimensionally, the higher the importance of a node.

The robustness of a network is its ability to maintain its function even after an attack or error [56]. In networks with exponential distribution, attacks and errors break the network into several small clusters. Scale-free networks with an inhomogeneous degree distribution, such as World Wide Web (W.W.W), and social networks, show a surprising degree of robustness to error. Even with high failure rates, the node’s communication ability can exist [39]. However, the robustness of networks lies in their high degree nodes. If such nodes are the center of the attack, then the network breaks, hence making the network vulnerable [4,5,39]. Attack and error strategies have been investigated in multilayer networks [40]. The authors in [56] report a comprehensive analysis of the tolerance to attack and error on seven different datasets. In [57], the authors investigate the robustness of multigraph scale-free networks to targeted attacks. A sequential reverse list of nodes is maintained based on degree. One sorts the nodes in degree-descending order. They also use the betweenness centrality, which counts the fraction of shortest paths in the network that goes through the node [39,57].

## 3. Proposed Methodology

Figure 1 presents the workflow diagram of the proposed analysis.

### 3.1. Data Description

GDELT [58] is a digital database containing news articles from around the world in several different languages. It is reachable through Google Cloud. The dataset of GDELT updates every 15 min. GDELT has four main classes, known as Quad class, namely 1 = Verbal Cooperation, 2 = Material Cooperation, 3 = Verbal Conflict, and 4 = Material Conflict. For example, if the Quad class is 4, both parties were involved in a material-type conflict. ‘Event Root Code’ is the root level category/class, and Event Code is the CAMEO (Conflict and Mediation Event Observations) code that falls under ‘Event Root Code’.

This work uses the ‘Mass Killing’ dataset of GDLET. It comes under the category ‘Material Conflicts’. ‘Event Root Code’ for ‘Mass Killing’ is (‘20’, Use Unconventional Mass Violence), and Event code is (202, ‘Engage in Mass Killing’). Genocide scholars define Mass Killing as the Killing of group members without the intention to eliminate the whole group. It also applies to Killing large numbers of people without clear group membership. In the GDELT source, “Mass Killing” refers to the Killing of multiple people without the intent of genocide. According to the description published by the GDELT source, it is primarily politically motivated. “Kurdish militia left two troops dead and 31 injured” is a typical example of such an event reported in GDELT. We retrieved data from the GDELT repository using the event code representing the single phrase “Mass Killing”. However, GDELT determines what “Mass Killing” means because it was a term retrieved in relevance to Mass Killing. GDELT news protagonists can be countries, international or regional militarized teams, or international actors. We filtered the data so that a source cannot be an NGO, group, or a non-state local contributor.

Each news event mention two actors. Actors are the interacting entities in the news. It reads as “Actor1 acted on Actor2”. Each news article is associated with several additional fields related to date, name, country, geography, code for the country, geography, Goldstein score showing the intensity of Conflict, quad class, root event code, and event code. Each event has a global event id.

The total number of events for Mass Killing (MK) equals 127,089. As we are interested in analyzing networks based on actors involved in MK, we extract the following attributes for each event: Global Event Id, SQL DATE, Actors code, and actors name. As actors with similar ‘actor codes’ are found in different locations to get a unique identifier for actors, we combine ‘ActorCode’ with ‘ActorGeo_ADM1Code’ using the separator ‘_’. After that, leading and trailing spaces that do not produce good results are removed. After the filtering process, the size of the dataset is reduced to 112,781 events.

The GDELT dataset contains multiple types of information, including the collection of news on the Mass Killings in the world. News has been collected around the world in the last 20 years. Each record in the news involves two actors that are contributing to the particular type of event. Every event/row has a link to where we may acquire the most up-to-date information. It also informs us of the date on which the event occurred. However, it does not provide information on how many people are killed in each occurrence or on the financial damages. Moreover, the Killing over religious grounds is also underrepresented in the GDELT database. However, utilizing the knowledge at hand can accomplish a great deal since we have identified people who are heavily involved in the spread of such events.

### 3.2. Network Construction

In this paper, a multigraph is considered to be the network structure. It is a graph with parallel edges and self-loops. A multigraph also called multiple or multivariate graphs, comprises many actors and a collection of relations that determine how actor pairs are linked [3,4]. We adopt the same network construction described in the papers [5,6]. It is illustrated in Figure 2 and the data is given in Table 1.
Any two actors involved in an event are linked with a unit weight of 1. The actors are called to be co-actors.For example, suppose an actor, let us presume A1, is involved in multiple events during a specified time duration. The node size is proportional to its number of events.

GDELT is a digital news dataset. There are news stories about various events happening worldwide on every specific date. Every record contains Actor1 and Actor2 and many more attributes related to the actors. Actors are the nodes that have a link with each other because they are quoted in the same record. Since it allows a multigraph network creation, one can find the exact relation between two nodes at various time stamps. The edge’s weight (thickness) depends on how frequently that link appeared in the dataset. Similarly, the edge thickness is proportional to the number of times an actor pair is involved in the events. The link between the nodes is called an actor pair link.

## 4. Results and Discussion

### 4.1. Quantitative Analysis of Network

The power-law behavior is typical in many situations. One can use it to depict a phenomenon where a probability of a small number of events is expected, while the likelihood of more significant events is uncommon. Unfortunately, understanding the power law is not as easy because of the fluctuation in the tail of the distribution. The power law is in the form: p(x) = K $X^{\alpha }$, where “α” is the scaling parameter [7,8,9]. It can be represented by the cumulative distribution function (CDF), the probability density function (PDF), or the complementary cumulative distribution function (CCDF). However, usually, one prefers CCDFs for envisioning a power-law heavy tail behavior [7,8,9]. If a distribution is plotted with logarithmic axes and shows a straight line, it demonstrates power-law distribution.

The investigation of the GDELT news is at the day level. Indeed, we analyze the number of news events reported every single day.

The number of events that occurs every single day is a random variable. The complementary cumulative probability distribution function (CCDF) for the number of events performed every day is shown in Figure 3. A power law with exponent 2.82 ± 0.025 is a good fit for these data. This paper also investigates the actors involved in MK. One can observe that certain actors are frequently engaged in such activities making their network bigger and bigger with respect to time, and some actors do not grow their network. One can inspect the network throughout other quantities such as interaction with actors, i.e., actor mention, actor pair (co-actor mention), and actor connected to other unique actors.

Figure 4 shows the CCDFs plots visualization of actor mention, actor-pair and actor degree for aggregated.

We analyze the network from three perspectives.
The actor mention: It is the number of times an actor is involved with other actors in inciting an event. It is denoted by m. Its complementary cumulative density function (CCDF) indicated by Qa(m) fits well a power law with exponents 1.67 ± 0.006,The actor-pair mention: It measures the number of times an actor-pair is involved in an event. It is denoted by w. Its complementary cumulative distribution function (CCDF), represented by Qab(w), fits well a power law with exponents of 1.48 ± 0.003.The number of unique actors’ k connected to a node: The complementary cumulative distribution function for actor degree is denoted by Qa(k). The probability of actor degree fits well a power-law with exponents 1.77 ± 0.007.

CCDFs plot for actor mention, actor pair mention, and actor degree are also visualized for each year in Figure 5. Plots are noisy because there is less data for each year than the aggregated data plots (2001–2020). However, it still seems that the power law is a good fit. The exponents’ values for the three quantities are reported in Table 2.

Power-law exponents with standard error sigma have been calculated using MLE [34,49,50,51,52].

In a random network, most nodes have comparable degrees and no hub. In contrast, in a network with a power-law degree distribution, there are numerous nodes with few links held together by a few highly connected hubs. Typically, in real-world networks, the power-law indices range between two and three. The network topology moves towards a hub and spoke configuration for low values of the exponent, while for high values, it looks like a random network.

We study the network from three perspectives: the actor mention in Figure 4a, the co-actor mention in Figure 4b, and the actor degree mention in Figure 4c on an aggregated basis. Results show that while most actors are comparatively less engaged, the statistics above quantitatively represent the variety in actor activity: The actor mentions Qa(m) power-law distributions show that a sizable minority are persistently involved in issues of “Mass Killing”. The power-law distributions for actor Mention Qab(w) suggest similar traits for actor pairs. The wide degree distributions Q(k) reveal the importance of the number of actors interacting with a vast number of actors.

Furthermore, we also investigated the power-law distribution every year for the three perspective actors mention in Figure 5a, the co-actor mention in Figure 5b, and the actor degree mention in Figure 5c on an aggregated basis. Results show that the counts are lower for the individual years due to less aggregation, which causes the data to be noisier. However, the degree distributions still have a power-law tail.

When we refer to Mass Killings, we do not use any information about the number of people concerned by the event. Consequently, as we interpret “Mass Killing” as a single word to extract data from the GDELT repository without any further analysis, there is no direct relation between the number of people involved in the event and the present study. However, one can note that the proposed framework can incorporate the number of people involved using a weighted network. The rsults of our investigations show no clear relationship between the various power-law indices of the different measures. It behaves as if each measurement is independent. The only consistent behavior is that they all share the property of obeying a power-law distribution.

### 4.2. Cluster Analysis

After inspecting and visualizing the network for the aggregated years (2001–7 July 2020), one can observe that the network consists of several separate components. It simply means that the actor involved in one event is never concerned about the other events throughout the duration. After inspecting the size of clusters, it appears that it grows linearly with the network size. The size of the largest cluster is 103–104, greater than the smaller cluster. Results are reported in Table 3 and visualized in Figure 6 (Cluster analysis using CCDF plots).

Figure 6 shows cluster analysis for each year. Power law fit for each cluster is in Table 4.

Table 3 reports the total numbers of nodes and the size of the largest component s1 commonly referred as the giant component for each year. Figure 7 illustrates the relation between these two quantities in a log–log scale.

We compute the variation of the size of the largest cluster with respect to the number of actors present in that cluster. First, one calculates the network’s size and the largest connected component for each year and the aggregated years (2001–2020). The power law fits the data with an exponent value of 1.12 ± 0.03. One can see in Figure 7 that the size of the largest cluster increases super linearly with the network size. Table 4 reports the power-law exponents.

### 4.3. Network Growth Properties

Computational models are valuable for testing theories and for inspecting activities over time. To examine influential actors that grow with respect to time, we extract the top 10 actors based on degree and mention parameters. We calculate cumulative growth rates π(m) for actor mention m and π(k) for actor degree. We calculate Π (x) = Δ x/Δ t to determine the growth rate for a given quantity x. We found that (x) is quite noisy in real data. Therefore, we computed the cumulative integral, $\int_{0}^{x}$Π ($x^{\prime }$) dx (x) which is less noisy. Growth has been calculated and analyzed by many authors on the datasets they used to calculate the growth pattern of various dynamics of the network [2,4,27,30].

Plots in Figure 8 suggest that π (m) and π (k) are superlinear functions of their argument degree and mention, respectively. Table 5 presents their exponent’s calculation. Plots in Figure 8 also show that actors’ growth is not independent of their respective arguments, i.e., degree and mention.

The plots in Figure 8 shows the growth pattern of top 10 actors based on their degree and mention.

### 4.4. Tolerance to Attack and Failure

To study the robustness of the ‘Mass Killing’ dataset, the network is set to break by targeted (Figure 9a) and by random (Figure 9b) removal of nodes [5,6]. Understanding how a network is robust to a targeted attack is essential in network science. Targeted attack experiments based on centrality measures proceed as follows.
Compute the largest connected component of the networkChoose a centrality measure accounting for a node’s importanceRank the nodes according to the centrality measure.Remove the nodes by descending order of centralityAfter removing the nodes found in the previous step, the process is repeated.

In Figure 9a, the behavior of the network under the degree targeted attack can be seen, the nodes with a high degree are removed, and the network breaks rapidly. The largest component breaks after the removal of less than 10% of nodes under degree targeted attack. In contrast, in random (Figure 9b) removal of nodes, as there is a large number of nodes with a low degree, the probability of selecting such nodes is very high, and the attack fails to break the network.

Attacks split the network into multiple components. “f” is the fraction of nodes removed, “G” is the fraction of nodes in the largest connected components and “c” refers to the average size of isolated clusters other than the giant component.

Under a targeted attack (Figure 9a), for a small value of “f”, the network fragments into clusters of varying sizes. However, there is still one largest connected component. As “f” increases, the clusters divide into single nodes or clusters of size two. Therefore the value of “c”, increases until “c” = 2. After that, it starts to decrease. Indeed, a small value of “f” only leads to the separation of single nodes and c ≃ 1. As “f” grows, the size of the pieces that break off the main cluster also grows, showing a unique behavior. When removing only 5% of the nodes from the network, the main cluster breaks into small pieces, leading to G ≃ 0, and the size of fragmentation c ≃ 2 is at its peak. As we continue to remove nodes, we fragment these isolated nodes, leading to a decreasing c.

Under random attack (Figure 9b), we do not observe a fragmentation point. Instead, the size of the largest cluster gradually shrinks. The network is deflating as a result of nodes breaking off one at a time, the rising error level leading to the isolation of single nodes, not clusters of nodes. On the other hand, when the network is broken under random attack, No fragmentation point is observed for random failures; instead, the size of the largest cluster gradually shrinks. The network is deflating as a result of nodes breaking off one at a time, the rising error level leading to the isolation of single nodes, not nodes of clusters.

For 20 years, the Figure 10 displays the percentage of top actors who have participated in the propagation of Mass Killing events. By establishing connections among the actors, the network of actors is developed. That network is processed through centrality analysis. It reveals several key players over 2 decades. Over the last 20 years, several countries have been deeply involved in Mass Killing events. Here, we attempt to break the network and identify those nodes that have made the network considerably stronger as time has passed by targeted attacks. The image above depicts the top recognized nodes with the highest network engagement, such as the United States (91.2%), France (87.2%), Israel (82.8%), Syria (78%), Iraq (51.2%), Palestine (49%), China (48.8%), Germany (44%), London (44.6%), and so on.

We discovered ways to disrupt the network of such mass violation events through testing, allowing government and policymakers to intervene and stop the spread of such events. As a result, rather than randomly eliminating nodes, we discovered that removing targeted actors (i.e., actors with a bigger number of neighbors are selected in every largest component) can break such a violating network.

The present study performs several investigations to unravel the hidden patterns of corporations and conflicts among the key actors during the global events of Mass Killings. It aims to understand the structural representation of corporations among countries during challenging times of Mass Killings and violence. The GDELT dataset is employed to investigate the interactions among the actors. These interactions are considered undirected graphs. Results show that the actor graph based on news events on Mass Killing exhibit a power-law degree distribution. It expresses that a few key actors are related to most of the global events of Mass Killing and violence. Moreover, one can also consider that the news data on global Mass Killings and violence focus mainly on a few actors. That might be possible because the havoc of Mass Killing concentrates in a few places worldwide. Further investigation reveals that the key players involved in the news about most of the Mass Killing events in the past 2 decades are the USA, Israel, and Syria. Further investigation shows that the USA is an active country trying to solve global issues. Consequently, it is involved in a majority of international events. In contrast, although Israel and Syria are actors of localized events, there exist several news events over the period that led the two countries to the top of the contributing actors. One of the main lessons learned from this analysis is that a few countries can negotiate and regulate to bring peace and minimize conflicts making a significant impact on the overall world peace.

## 5. Conclusions

News reports in media play an essential role in bringing out news analysis. This work also uses the digital news presented in the GDELT dataset for a specified time duration. The aggregated data over the years makes it possible to build a network and visualize and inspect it. We study the network from three perspectives: actor occurrences, co-actor occurrences, and the degree of a node. These are good representatives of the intensity, impact, and significance of the engagement of actors with others. This study shows that there are large clusters. In other words, most actors form a giant component throughout time and hence are influential. There are also tiny clusters. Indeed, some actors do not grow their network over time and are not influential. Identifying such actors frequently involved in spreading such conflicting events is a critical approach that can help policymakers such as organizations, governments, etc., make their policies and help prevent the spread of such events. On the basis of centrality measures, the United States, France, Israel, Syria, Iraq, Palestine, China, Germany, and London were identified as the top nodes deeply involved in Mass Killings over a 20-year period. The probability distribution of actor mention, co-actor mention, and actor degree have a power-law decay in their tails, revealing the underlying mechanism behind the events. The contribution of this work is to identify, investigate and formulate a time-variant conflicting complex network concerning the actor’s involvement in the global news event. This process is not restricted to the global news dataset but can also apply to other datasets. The mixed approach used here offers a sufficient understanding for analyzing such complex networks.

## Figures and Tables

**Figure 1 entropy-24-01017-f001:**
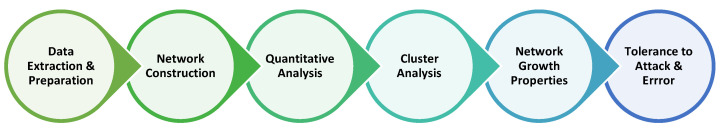
Flow of the proposed work.

**Figure 2 entropy-24-01017-f002:**
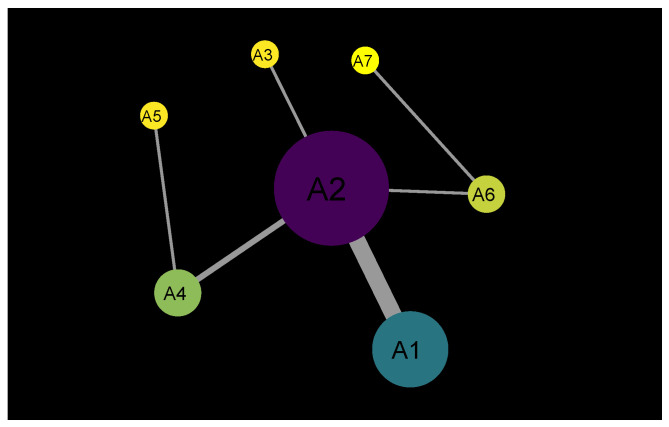
Sample network visualization (data from Table 1). It is the most straightforward visualization to understand how real complex network is built. The size of a node increases as the number of times it appears in news events increases. The weight is the number of mentions/occurrences of an actor pair in the news. The more often a specific edge is involved in events, the more prominent it becomes.

**Figure 3 entropy-24-01017-f003:**
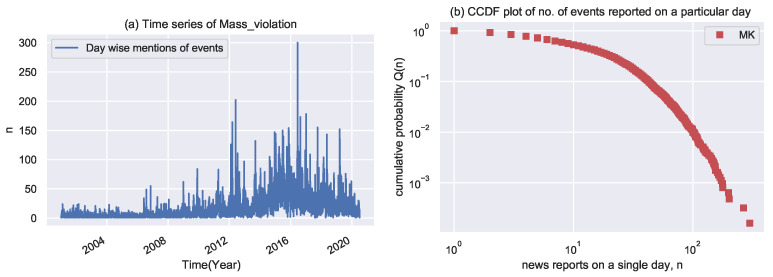
Visualization of no. of events every single day. (**a**) Time series of no. of events reported on a single day, for ‘Mass Killing’ dataset, and (**b**) shows CCDF Q(n) that n or more events happened on a particular day. Data seems to fit power-law decay with exponents 2.82 ± 0.025.

**Figure 4 entropy-24-01017-f004:**
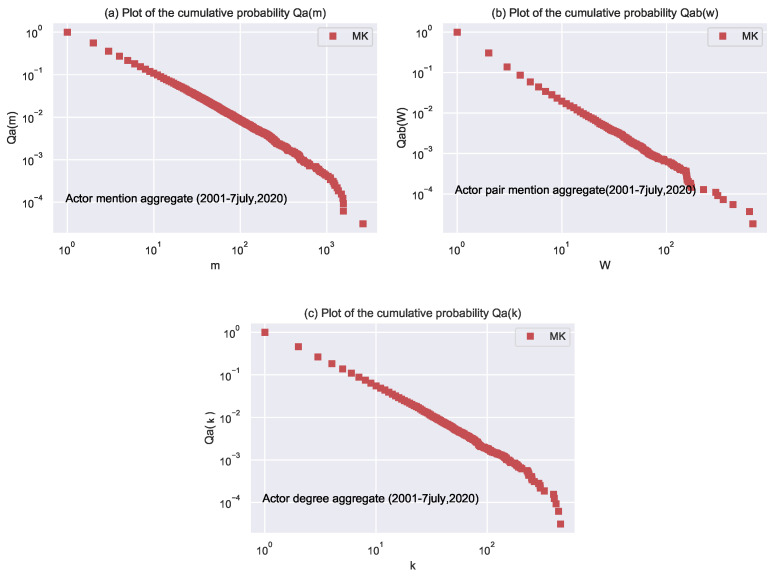
The figure shows the complementary cumulative distribution function (CCDF) for aggregated years for the following parameters. (**a**) Plot of CCDF for actor Mention Qa(m) aggregate that an actor is mentioned m or more times follows a power-law distribution with exponents 1.67 ± 0.006. (**b**) Plot CCDF for actor pair mention aggregate Qab(w) indicates that an actor pair is mentioned w or more times which follows a power-law distribution with exponents 1.48 ± 0.003. (**c**) CCDF for actor degree aggregate Qa(k) indicates that an actor is co-mentioned with k or more actors following a power-law distribution with exponents 1.77 ± 0.007.

**Figure 5 entropy-24-01017-f005:**
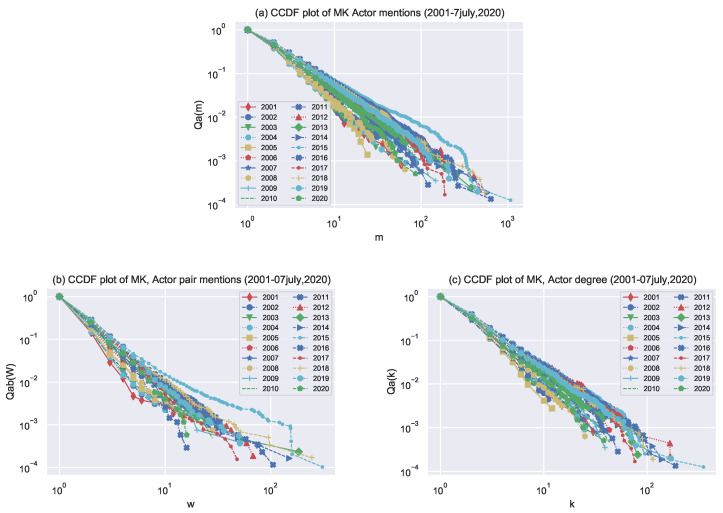
Plot of cumulative probability CCDF Qa(m) an actor is mentioned m times or more, Qab(w) indicates that an actor pair is mentioned w or more times, Qa(k) indicates that an actor is co-mentioned with k or more actors. Because there is less data for each year than in the aggregated years (2001–2020), the graphs are noisy. However, the power law appears to be a suitable fit. The actual fits are given in Table 2.

**Figure 6 entropy-24-01017-f006:**
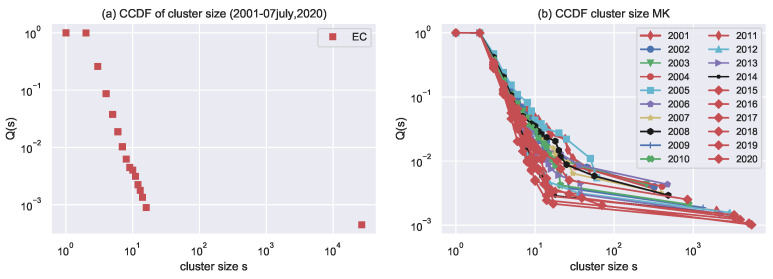
Cumulative probability CCDF charts are used to analyse clusters. (**a**) For aggregated years, a CCDF Q(s) plot of a cluster of sizes higher than s. (2001–7 July 2020). In comparison to the others, the largest cluster is quite enormous. (**b**) The CCDF Q(s) plot demonstrates that there is a cluster of size greater than s.

**Figure 7 entropy-24-01017-f007:**
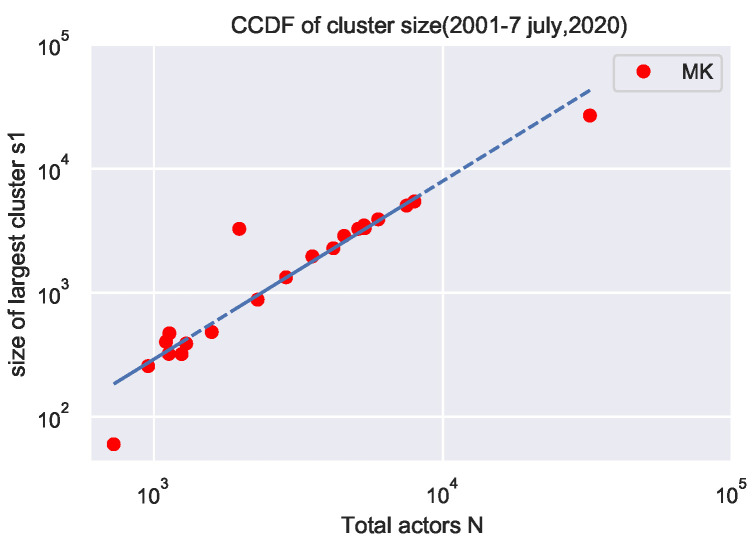
Variation of the size of the largest cluster s1 with network size N, for different years (2001–2020) as well as for aggregate years (2001–2020).

**Figure 8 entropy-24-01017-f008:**
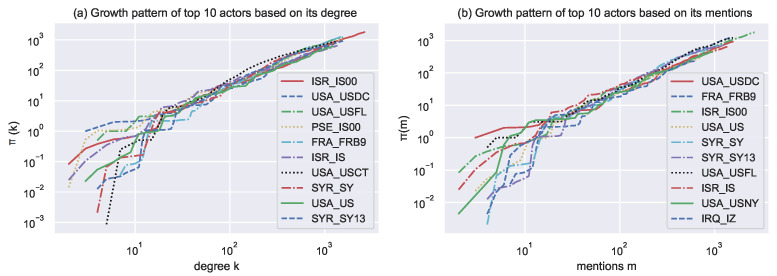
Cumulative growth rates for Actor mention π(m) and for Actor degree π(k) for the top 10 actors based on their mention and degree, respectively. The plot shows that π(m) and π(k) are super linear functions of their respective arguments, degree and mention, respectively. Fitting exponents are given in Table 5.

**Figure 9 entropy-24-01017-f009:**
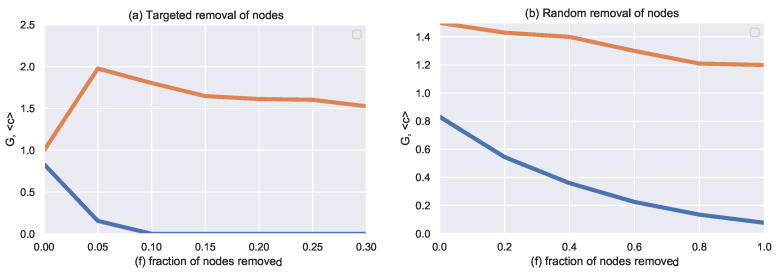
The network’s structure under degree targeted attack: One removes nodes in descending order of their degree. The plot depicts the evolution of the giant component G (fraction of nodes in the largest connected component) and the average number of nodes in clusters other than the giant component (c), with an increasing percentage of nodes removed (f). Targeted node removal (**a**) is more effective at splitting the network than random removal (**b**). The results are for the networks aggregated over 2001–2020.

**Figure 10 entropy-24-01017-f010:**
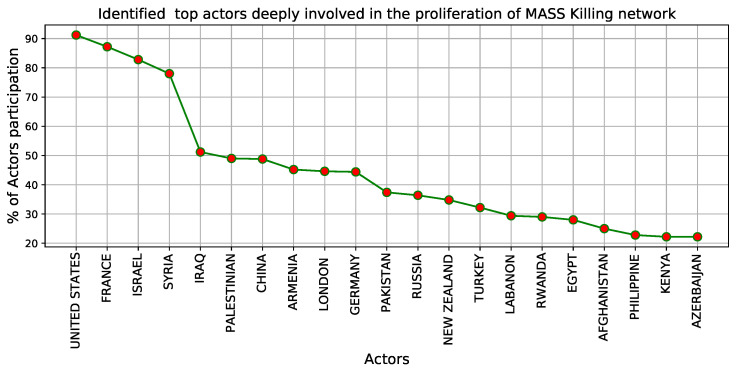
For 20 years, this graph displays the percentage of top actors who have participated in the propagation of Mass Killing events.

**Table 1 entropy-24-01017-t001:** Sample data for the network construction (illustrated in Figure 2).

Events	Node1	Node2
E1	A1	A2
E2	A2	A3
E3	A4	A5
E4	A1	A2
E5	A2	A4
E6	A6	A7
E7	A1	A2
E8	A1	A2
E9	A1	A2
E10	A1	A2
E11	A4	A2
E12	A2	A6

**Table 2 entropy-24-01017-t002:** Exponents calculations for actor mention, actor pair mention and actor degree for individual and aggregated years (2001–7 July 2020).

Year	Actor Pair Mention	Actor Mention	Actor Degree
2001	1.29 ± 0.01	1.71 ± 0.02	1.56 ± 0.02
2002	1.27± 0.01	1.73 ± 0.02	1.52 ± 0.02
2003	1.25 ± 0.01	1.62 ± 0.02	1.53± 0.02
2004	1.24 ± 0.02	1.68 ± 0.02	1.53 ± 0.02
2005	1.29 ± 0.01	1.85 ± 0.03	1.61 ± 0.02
2006	1.32 ± 0.01	1.77 ± 0.02	1.57 ± 0.02
2007	1.49 ± 0.02	1.91 ± 0.03	1.45 ± 0.01
2008	1.34 ± 0.01	1.88 ± 0.02	1.62 ± 0.02
2009	1.31 ± 0.01	1.91 ± 0.02	1.64 ± 0.01
2010	1.38 ± 0.01	1.79 ± 0.02	1.57 ± 0.01
2011	1.35 ± 0.01	1.73 ± 0.01	1.6 ± 0.01
2012	1.43 ± 0.01	1.89 ± 0.01	1.71 ± 0.01
2013	1.45 ± 0.01	1.87 ± 0.01	1.61 ± 0.01
2014	1.47 ± 0.01	1.92 ± 0.02	1.65 ± 0.01
2015	1.5 ± 0.1	1.96 ± 0.01	1.66 ± 0.01
2016	1.48 ± 0.01	1.95 ± 0.01	1.65 ± 0.01
2017	1.44 ± 0.01	1.91 ± 0.01	1.63 ± 0.01
2018	1.48 ± 0.01	1.93 ± 0.02	1.63 ± 0.01
2019	1.45 ± 0.01	1.91 ± 0.01	1.62 ± 0.01
2020	1.42 ± 0.01	1.82 ± 0.02	1.51 ± 0.01
2001–2020	1.48 ± 0.0	1.67 ± 0	1.48 ± 0.0

**Table 3 entropy-24-01017-t003:** Shows statistics of total number of nodes and total nodes in each year’s largest component.

Year	Total Nodes	Largest Cluster Size
2001	1247	320
2002	1126	321
2003	955	256
2004	1101	402
2005	726	60
2006	1131	471
2007	1293	389
2008	1584	482
2009	2865	1331
2010	2284	879
2011	3532	1963
2012	4551	2871
2013	4175	2282
2014	5330	3486
2015	7964	5446
2016	7482	5040
2017	5970	3906
2018	5359	3315
2019	5093	3275
2020	1973	843
2001–2020	32,213	26,796

**Table 4 entropy-24-01017-t004:** Power-law exponents for cluster analysis for individual years.

Year	Exponent	Year	Exponent
2001	1.98 ± 0.06	2011	1.85 ± 0.02
2002	1.82 ± 0.05	2012	1.85 ± 0.02
2003	1.83 ± 0.05	2013	2.03 ± 0.02
2004	1.75 ± 0.04	2014	1.88 ± 0.02
2005	1.6 ± 0.08	2015	1.9 ± 0.01
2006	1.86 ± 0.04	2016	1.87 ± 0.01
2007	1.69 ± 0.04	2017	1.99 ± 0.02
2008	1.94 ± 0.05	2018	1.88 ± 0.02
2009	1.96 ± 0.03	2019	1.96 ± 0.02
2010	1.85 ± 0.03	2020	1.81 ± 0.03

**Table 5 entropy-24-01017-t005:** Exponents calculations for the top 10 actors based on degree and mention.

Actor	Actor Degree	Actor	Actor Mention
USA_USDC	2.36 ± 0.61	USA_USDC	2.36 ± 0.61
FRA_FRB9	2.53 ± 0.41	FRA_FRB9	2.53 ± 0.41
ISR_IS00	3.02 ± 0.83	ISR_IS00	3.02 ± 0.83
USA_US	3.09 ± 0.94	USA_US	3.09 ± 0.94
SYR_SY	2.13 ± 0.36	SYR_SY	2.13 ± 0.36
SYR_SY13	1.89 ± 0.28	SYR_SY13	1.89 ± 0.28
USA_USFL	2.33 ± 0.47	USA_USFL	2.33 ± 0.47
ISR_IS	2.09 ± 0.55	ISR_IS	2.09 ± 0.55
USA_USCT	1.67 ± 0.21	USA_USNY	2.3 ± 065
PSE_IS00	2.81 ± 0.81	IRQ_IZ	1.99 ± 0.49

## Data Availability

The dataset of GDELT has been used for research purposes and it is available online. It can be accessed using Google’s Big Query and is available here: https://www.gdeltproject.org (accessed on 1 March 2021).

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
