# Peer review of "Complex Network Analysis of Mass Violation, Specifically Mass Killing"

_entropy, 2022, doi:10.3390/e24081017_

Round 1

Reviewer 1 Report

This is an interesting manuscript analysing digital datasets (GDELT) of events in media with the aim of gaining insights into the events and involved actors. I find the results of the power-law distributions of different measures (actor mention, co-actor mention, actor degree, etc) and time-dependent analysis quite interesting and would recommend the publication of this manuscript if the authors address my comments below.

1. The focus of the reported analysis is "mass" killing, "mass" violence, etc. However, there isn't a unique definition of the term of "mass", I would think. Please define what "mass" means.

2. Related to the point above, I would think that the results (e.g. power-law indices) might depend on the definition of "mass". Have the authors looked into this?

3. It is interesting to see the power-laws on short and long time scales. However, having found those power-laws, the authors should provide some insights on their meanings throughout the manuscript. For instance, what do large and small power-law indices mean?

4. Also, what would be relations among different power-laws indices of different measures?

5. There seem to be quite many places in the text where figure numbers are not correctly referred to. I suggest the authors to go through the text carefully to make sure that correct figure numbers are used.

6. Figures: figure labels should be carefully checked and corrected (there are missing labels, etc, etc). For instance, what does "(f) fraction of nodes remove" mean on x-axis in Figure 10. (why is there "(f)"?)

7. Figure and table citations in the text should be checked throughout the text and should be corrected to match figure numbers with the actual figures that are referred to. For instance,

i) on Line 320, "Figure (a)" -> "Figure 10 (a)"; "Figure (b)" -> "Figure 10(b)" 

[There seem to be other typos on this line, so please check]

ii) on Line 330: "Figure below" -> "Figure (11)"

etc, etc.

8. There are numerous minor typos (esp, in "spacing", "comma"). For instance,

i) on Line 36: "[9].In" -> "[9]. In"

ii) on Line 83 and many other places: similar spacing problem

.....

iii) on Line 336: "shows shows" -> "shows"

Reviewer 2 Report

The authors apply complex network analysis to the max killing problem. They extract the data from GDELT database. They built the network considering the events and the news correlated with them. They studied the power-law parameter alfa and its evolution during the years, detect influencers, and how to damage the network of influencers for avoiding their influence.

The results are very interesting and explained in a synthetic and clear way. The paper deserves to be published in its present form. I have only minor comments

Explain the acronym the first time GDELT is introduced

Page 9:  Figure 5 has to be Figure 6

Page 10: Table 4 has to be changed in Table 3

The caption of Figure 7 seems incomplete, it finishes with “then”

Pag 12:  explain the meaning of commutative growth

Page 14:  “it can be seen in Figure Below”, has to be changed in “it can be seen in Figure 10”,

In the same paragraph, there is the repetition of a sentence

On page 15, in the first paragraph the citation of Figure 11 is missing

Round 2

Reviewer 1 Report

I find the authors' response and revision satisfactory and am happy to recommend it for publication in Entropy.